# The Exciting Realities and Possibilities of iPS-Derived Cardiomyocytes

**DOI:** 10.3390/bioengineering10020237

**Published:** 2023-02-10

**Authors:** Fuga Takahashi, Praneel Patel, Takahiro Kitsuka, Kenichi Arai

**Affiliations:** Department of Clinical Biomaterial Applied Science, Faculty of Medicine, University of Toyama, Toyama 930-8555, Japan

**Keywords:** iPS-derived cardiomyocytes, cardiovascular disease, embryonic stem cells, Wnt signaling, pluripotency

## Abstract

Induced pluripotent stem cells (iPSCs) have become a prevalent topic after their discovery, advertised as an ethical alternative to embryonic stem cells (ESCs). Due to their ability to differentiate into several kinds of cells, including cardiomyocytes, researchers quickly realized the potential for differentiated cardiomyocytes to be used in the treatment of heart failure, a research area with few alternatives. This paper discusses the differentiation process for human iPSC-derived cardiomyocytes and the possible applications of said cells while answering some questions regarding ethical issues.

## 1. Introduction

Cardiovascular disease (CVD) affects a concerning number of patients across the globe, accounting for 17,921,000 deaths as of 2017 [1]. Some of those CVD-related deaths can be attributed to intrinsic factors such as long QT syndrome, where a mutation in the gene encoding a sodium ion channel leads to unexpected arrhythmias [2]. Cardiac injury, such as a myocardial infarction (MI), deprives cardiac muscles of oxygen and causes them to perish; the human heart fails to regenerate cardiomyocytes and instead develops intense scarring that reduces contractibility and blood flow in the heart [3,4]. A study showed how the regenerative capacity of the adult heart declines rapidly after birth, with less than 1% of the heart’s cardiomyocytes being replaced each year [5]. Subsequent heart attacks can prove deadly to patients, and heart transplantation offers the best survival benefit for said patients with end-stage heart failure. However, rejection, a shortage of heart donors, and limited graft availability limit this treatment option [6]. Other treatment options include pharmacological treatments such as beta-blockers, angiotensin receptor blockers, I(f) inhibitors, and more [7]. A study by Teo et al. compromising 53,268 patients in 55 trials suggested a 19% reduced mortality rate after myocardial infarction with the use of beta blockers [8]. However, the clinical guidelines for pharmacological treatment of myocardial infarction are outdated. Goldberger et al. found that out of 3000 patients in OBTAIN (outcomes of beta-blocker therapy after myocardial infarction), the lowest mortality rates were found with beta-blocker intakes of 12.5–25% of the recommended doses [9].

In 2006, Takahashi and Yamanaka successfully reprogrammed somatic cells into pluripotent stem cells by introducing four factors, Oct3/4, Sox2, c-Myc, and Klf4, under ES cell culture conditions through the use of a retrovirus [10,11]. Called iPSCs, the cells became a preferred alternative for ESCs, cells discovered in 1952 when Briggs and King conducted somatic cell nuclear transfer experiments in which the nucleus of a somatic cell was transferred to the cytoplasm of an enucleated cell [12].

One of the most promising differentiation routes includes human iPSC-derived cardiomyocytes. The ability to create cardiomyocyte tissue, which induces contractile motion in the heart, opens new avenues of regenerative research such as transplantation, bioprinting, disease modeling, and cardiac patches (Figure 1) [13,14].

## 2. Differentiation of iPSC-Derived Cardiomyocytes

Human iPSCs can be differentiated into cells from the following 3 germ layers: the ectoderm, which includes astrocytes, epidermal cells, keratinocytes, and hair cells; the endoderm, which include hepatocytes, lung alveolar cells, and intestinal epithelial cells; and the mesoderm, the category that includes cardiomyocytes [16]. Differentiation of cardiomyocytes from human iPSCs is a necessity for regenerative treatment and must be pursued further since adult cardiomyocytes are terminally differentiated, unable to proliferate independently, and are permanently withdrawn from the cell cycle [17]. For this reason, differentiation from human iPSCs, aside from differentiation from human ESCs, proves to be one of the very few ways for patients to increase their number of cardiomyocytes [18]. Due to the increasing amount of ethical questions raised by use of ESCs, an alternative must be found.

The perfect combination of external and internal stimuli, such as the variety of molecules and growth factors, must be found for iPSC-derived cardiomyocyte differentiation. It is also important to note that cardiomyocytes can be differentiated into atrial or ventricular cells; the differentiation process, gene expression patterns, structural proteins, ion channels, and ultrastructure vary between the two types of cells [19,20]. Devalla et al. found that in retinoic acid-treated cells, upregulation of atrial markers such as SLN, HEYL, PITX2, and NPPA led to the downregulation of ventricular markers such as MYL2, IRX4, HAND1, and HEY2 [21]. The number of variable factors that can change with iPSC differentiation also adds to the complexity and absence of uniformity needed for clinical therapeutic applications [22]. In this section, we point out a variety of studies with positive results to help steer the medical community in a similar direction.

### 2.1. iPSC and ESC Differentiation—What Changes?

Different protocols have been created for the differentiation of human iPSC-derived cardiomyocytes, which have been adapted from ESC protocols. In both types of differentiation, establishing proper combinations of soluble signals that trigger the proper signaling pathways (Activin/Nodal, Wnt, BMP) is imperative for cardiomyocyte specification [23].

Zhang et al. conducted a study with iPS clones of fetal and newborn origin (reprogrammed with lentiviral-mediated transduction of OCT4, SOX2, NANOG, and LIN28) as well as human ESC lines H1 and H9 [24,25,26]. After maintaining the cells in mouse embryonic fibroblasts, supplementing the cells with zebrafish fibroblast growth factors, incubation, and then suspension, the team found that ESC-derived cardiomyocytes (H1 and H9) and their iPSC-derived cardiomyocytes (IMR90 and Foreskin C1) did not have major differences in action potential and number of spontaneously beating cells [25].

Moreover, Zhao et al. found very few differences between the cardiac differentiation of human iPSCs, nuclear transfer ESCs (nt-ESCs), and in-vitro fertilization ESCs (IVF-ESCs). The team differentiated the cells into cardiomyocytes with a small molecule-mediated differentiation protocol, creating a sheet-like structure that beat for over 2 min, then compared the results [27]. Cardiac differentiation efficiency, expression of cardiac structure genes, expression of the atrial marker genes (NPPA, MYL7), and expression of ventricular marker genes (NPPb, MYL2) all showed very small differences between the three types of cells [27]. Gene expression heterogeneity was studied by qPCR and showed similar expression of cardiac structure genes (TNNT2, MYH7), gap junction genes (GJA1, GJA5), and vascular endothelial lineage genes (CD31, CD144) [27].

These trials may suggest that iPSCs could be a similar alternative to ESCs and replace them for cardiac regenerative medicine applications. However, due to the amount of contradicting studies (such as the study by Chin et al. in which 3947 expression differences (out of 17,620 genes) were claimed to have been found between human-ESC and early passage human-iPS cell lines, suggesting that the human-iPS lines did not silence the expression patterns of their somatic counterparts) in the field, more research must be completed to answer this question [18,28].

### 2.2. Embryoid Body (EB) Differentiation

EBs are spherical, small aggregates formed in suspension by PSC; the culture of EBs leads to the creation of specific cell lineages that allow cells to differentiate into cardiomyocytes. Although the effort is time-consuming, labor-intensive, and lacks monitoring for differentiation of EBs, it should be noted that a bioreactor reactor culture system with the ability to monitor temperature, pH, and oxygen levels would address such problems [29]. On the other hand, variations in differentiation protocols and experiment variables have led to inconsistent results about the number of beating and non-beating cardiomyocytes in EBs; the range of beating cardiomyocytes can range from 5–70% [18].

### 2.3. Wnt Signaling in Monolayer Differentiation

Monolayer differentiation consists of manipulating the Wnt signals (secreted factors that regulate cell growth, motility, and differentiation during embryonic development). First of all, glycogen synthase kinase 3 (GSK3) and it’s phosphorylation of β-catenin is inhibited by CHIR99021 and 6-bromoindirubin (BIO) in Wnt signaling, thus accumulating β-catenin and inducing the expression of mesodermal markers such as Brachyury (Bry) and Eomesodermin (EOMES) [30,31] (Figure 2). This allows activation of the primary cardiac mesodermal regulator, mesoderm posterior BHLH transcription factor 1(Mesp1), thereby causing inhibition of Wnt mediated by Dickkopf-related protein 1 and small molecules such as IWP4, IWP2, IWR1, XAV939, and WNT-C59 [32,33,34,35,36]. The whole process facilitates the specification of IPS cells to a cardiac lineage. The differentiation protocols that utilize Wnt signaling have several benefits that have attracted the attention of many scientists.

Monolayer differentiation has the ability to produce cardiomyocytes with high purity (over 80%), whilst allowing for large-scale differentiation of cardiomyocytes [34,38]. However, GSK3 inhibitor concentrations as well as culture periods must be optimized for a high cardiomyocyte yield. Zhao et al. found that varying amounts of maintenance CHIR doses–more specifically 2 μM–led to a significantly larger amount of cardiomyocytes [36].

A study in 2019 by Hamad et al. took advantage of Wnt signaling and CHIR to convert hiPSCs into mesodermal cells, then used IWP2 and XAV939 to differentiate the cells into cardiomyocyte progenitors with the simultaneous inhibition of the porcupine and tankyrase pathways [39]. Cardiomyocyte yields amounted to 0.5 million cells per cm^2^ of growth area [39]. It is also important to note that the addition of IWP2 and XAV939 during days 3–5 of cardiac differentiation and continuous supplementation of ascorbate increased the percentage of TNNT2-positive cells (cardiac troponin T-positive cells containing highly organized sarcomeres) to 90.2 ± 2.2%, clearly showing the reduced batch-by-batch variations of the cardiomyocytes [39].

#### 2.3.1. GSK3 Inhibitors in Wnt Signaling

Lian et al. created the 19-9-11 human iPSC line integrated with a lentiviral 7TGP vector to express green fluorescent protein (GFP) for immunofluorescent analysis. The CHIR-induced GFP-expressing cells expressed both Nkx2.5 (expressed in committed cardiomyocytes) and Is11 (Is11+ heart cells are capable of self-renewal and expansion before differentiation), demonstrating that CHIR induced differentiation with the mTeSR1 medium that was used [34]. The team also found that the GSK3 inhibitor still relied on β-catenin in the shcat-2 lines carrying the shRNA that downregulated β-catenin expression, showing a very small amount of Is11+ cells compared to the number of 19-9-11 human iPSCs.

#### 2.3.2. Only Moderating Regulatory Elements of Wnt Signaling

Lian et al., mentioned above, also tested Wnt signaling without any growth factors. The team used a β-catenin-knockdown hPSC line added dox after 36 h, and still created 98% cardiac troponin T-labeled cells (cTnT cells) [34]. In addition to immunostaining showing Is11+ and Nkx2-5+ cells, the line showed the expression of TBX5, MEF2C, and GATA4, which are important regulators of cardiomyocyte development, for the full length of the experiment [31,40,41,42,43].

#### 2.3.3. Other Factors in Wnt Signaling

Cell confluency before culture may also be important [44]. A study by Balafkan et al. used two hiPSC lines, Detroit 551-A and AG05836B-15, finding that the 60–70% range of confluency (out of 30–40%, 60–70%, and 80–90%) resulted in the culture with the highest number of functional cardiomyocytes [24,45,46,47,48].

Understanding the timing of Wnt signaling activation during varying embryonic developmental stages has also proven to be important to the successful culture of cardiomyocytes. Suppressing Wnt signaling at the beginning of differentiation proved to be very ineffective because Wnt signaling is stimulated and therefore most active in that time period before it is inhibited after mesodermal growth [49].

## 3. Enhancing iPSC-Derived Cardiomyocytes

In current academia, there is a shortage of research on the effects of human iPSC-derived cardiomyocytes in clinical trials [50]. As part of the “bench to bedside” approach investigators and regulatory agencies are taking in the development of the promising iPSC technology, Liu et al [51] noted that stem cell research has recently shifted toward clinical trials, with four landmark clinical trials since 2016. The safety of human iPSC-derived cells is paramount, especially in the case of human iPSC-derived cardiomyocytes, which are responsible for contraction of the heart, and the heart would fail without them [52]. However, given the large variety of applications for human iPSC-derived cardiomyocytes in regenerative and transplant medicine, disease modeling, drug screening, and more, the capacity for labs to select, differentiate, and maturate working iPSC-derived cells is important alongside the benefits the cells have to offer [53,54,55,56]. Currently, only small-scale production of iPSC-derived cells is possible, leading to a lack of stable, high-output production of iPSC-derived cells [57]. Efforts have been undertaken to enhance iPSC-derived cell production by a variety of researchers, largely focusing on enhancing pluripotency and maturation.

### 3.1. Pluripotency

Pluripotency refers to the ability of a cell to differentiate into one of the three primary germ cell layers, as listed in the introduction of Section 2. This is in contrast to totipotent cells, which can differentiate into any cell type, and multipotent cells, which can differentiate into a relatively limited variety of cells [58] (Figure 3).

Cardiomyocytes are part of the group of cells that can be differentiated from iPSCs. The trait of pluripotency is key to the function and optimism around iPSCs, as pluripotency allows for any cell to be in a stem-cell-like state capable of differentiating into another type of cell. Current challenges in inducing pluripotency include the difficulty of inducing pluripotency using solely chemical means and reprogramming with a small (<1 × 10^6^) batch of cells [59].

#### 3.1.1. Genetic Modifications

The genetic makeup of a cell can be manipulated to induce pluripotency using DNA methylation. Nishi et al. found that peptidylprolyl isomerase PIN1 modulated phosphorylation signaling, thereby regulating the expression of key genes in pluripotency, primarily OCT4 [60]. Another method of generating iPSCs–other than Wnt signaling–is utilizing extracellular signal-regulated kinase (ERK) and mitogen-activated protein kinase (MAPK) signaling, which is a relatively imprecise method. ERK has been shown to support pluripotency in mouse stem cells, but inhibition of MEK/ERK using MEK inhibitors also had the same effect [61]. There is evidence suggesting ERK signaling shifts a cell from a self-renewing, pluripotent state to a lineage-obliged state. By using MEK inhibitors to prevent fate determination, cells could be held in a self-renewing state, i.e., pluripotency [62]. However, this was in direct contradiction to prior evidence showing ERK increased pluripotency. Ma et al. hypothesized that a minimum level of ERK/MAPK may exist where stem cell proliferation, cell cycle progression, suppression of apoptosis, telomere length maintenance, and genomic stability are maintained above said level [61]. However, ERK may suppress self-renewal and develop a cell in tandem, which may explain the contradictory lab results of ERK/MAPK.

#### 3.1.2. Chemical Modifications

Growth factors and the associated kinases regulating intracellular signaling pathways during differentiation and pluripotency are targets for a chemical approach to induce pluripotency [63]. The chemical approach was developed to increase the reprogramming efficiency achieved using only genetic approaches [59]. A prominent method of increasing the pluripotency of iPSCs is by leveraging small molecule transcription factors to reprogram cells into pluripotency. When reprogramming cells, certain chemicals can be used to replace or support traditional OSKM reprogramming factors [63]. A variety of said chemicals have been discovered. Hou et al. first published a proof-of-principle 7-chemical combination (VPA, CHIR99021, E616452, Tranylcypromine, Forskolin, 3-deazaneplanocin A, and TTNPB) that enhanced OSKM factors. Among the most effective combinations was AM580, EPZ004777, SGC0946, and 5-aza-2-deoxycytidine, discovered by Zhao et al., which yielded a 1000-fold greater reprogramming than previously tested [64].

### 3.2. Maturation

iPSC cardiomyocyte maturation is a complex and intensive process, in which expression of sarcomeric genes is restricted, structural changes occur, and metabolism switches from glycolysis to oxidative phosphorylation [65,66,67]. To date, numerous methods have been investigated to improve the maturation process in both temporal and quality domains.

#### 3.2.1. Inhibition of MAPK and PI3K/AKT Pathways

Similar to inducing pluripotency, genetic changes can enhance the maturation process. Garay et al. published a study detailing the numerous ways in which human iPSC differentiation into cardiomyocytes was facilitated by inhibiting the MAPK and P13K/AKT pathways [65]. The inhibition of the two pathways improved the maturity of cells in a variety of domains, including hypertrophy, multinucleation, metabolism, T-tubule density, calcium handling, and electrophysiology. In terms of maturity, the improvements made the 30-day human iPSC-derived cardiomyocytes similar to the 60-day cardiomyocytes. 

#### 3.2.2. Cellular Shape and Environment

As iPSC-derived cells, including cardiomyocytes, are most often grown ex vivo, the environment in which cardiomyocytes mature is of the utmost importance to the future success of the cells. Jimenez-Vazquez conducted a study developing a microstructured silicone membrane acting as a cell culture substrate, with properties similar to the in vivo cardiac matrix [68]. By mimicking the natural environment of cardiomyocytes, Jimenez-Vazquez expected cells to develop with in vivo-like morphology—a technique that can be used to grow higher-quality cardiomyocytes in the future.

#### 3.2.3. Electrical Stimulation

Electric stimulation refers to the stimulation of transcription factors by electricity. In cardiomyocytes, electrical stimulation is linked to the triggering of NRF-1, GATA4, NFAT3 transcription factors and cytochrome C [69]. A separate study also showed that cardiomyocytes in neonatal rats directly upregulated Adss1 expression in muscle in proportion to the pacing of shocks [70]. Both studies alluded to the use of electroshock in being able to easily manipulate maturation at a rate that was easily controlled by an investigator, providing researchers with a large degree of testing capability.

#### 3.2.4. Fatty Acid Metabolism

Activation of Ppargc1a/b, Ppara, Nrf1/2, and Esrra/b/g–metabolic transcriptional regulators lead to an increase in fatty acid metabolism [71]. Riquelme et al. found that cardiomyocytes from pythons underwent hypertrophy in response to fatty acid treatments, which also stimulated the growth of the heart. It was also found that fatty acid treatment of hPSC-cardiomyocytes increased the cell area by 59%, increased the sarcomere length from ~1.73 μM to ~1.80 μM, and decreased the circulatory index from 0.63 to 0.58, showing that the treatment enhanced the structural maturation of the cell. Moreover, the same team found that the fatty acid treated-hPSC-cardiomyocytes had a ~3.3 nN/cell increase in twitch force–or contractile force–than the control cardiomyocytes [72].

#### 3.2.5. Epigenetic Priming

Polyinosinic-polycytidylic acid (pIC), a double-strand RNA that acts as an innate immunity activator, was found to accelerate the reprogramming of skin fibroblasts and yielded cardiomyocytes with increased maturity by utilizing epigenetic priming to enhance the expression of cardiac myofilament genes and Notch signaling [71]. Biermann et al. created hPSC-cardiomyocytes with a Wnt signaling protocol, adding pIC on days 3–5 of the protocol. It was found that pIC treatment almost doubled the size of the hPSC-cardiomyocytes; however, replating the cardiomyocytes on Matrigel patterned on a soft PDMS substrate was necessary to create the rod-like morphology of adult cardiomyocytes [73].

## 4. Applications of iPSC-Derived Cardiomyocytes

As listed in the introduction, researchers are exploring options for regeneration of the heart and cardiomyocytes. Cardiomyocytes can be injected into the body to increase their numbers, but they can also be used to create cell sheets to treat myocardial infarction. Moreover, iPSC-derived cardiomyocytes have potential in cardiac disease modeling, an area of research where current models are unreliable.

### 4.1. Transplanting the Cardiomyocytes

In clinical settings, human iPSC-derived cardiomyocytes are useful in a patient’s body. As many genetic and chemical procedures to induce pluripotency and mature cardiomyocytes rely on ex vivo conditions, cardiomyocytes must then be transplanted in the body. There are several approaches for transplantation. However, no matter the approach, transplantation must strive toward efficient retention, successful engraftment, and no immunological reaction.

Direct injection is an iPSC transplantation method in which a transplantation device is used to inject the material, usually cardiac spheroids with cardiomyocytes, into the myocardial layers [74]. A variety of devices have been developed to assist with direct injection. On such device is approaching clinical usage but falls short of immunological acceptance and arrhythmia [75]. Another device focuses on the distribution of cardiomyocytes within the heart and is geared for clinical applications in patients with heart failure [74]. Ultimately, with the range of solutions available, clinical application is the next step for direct injection technology after immunological and retention issues have been solved.

### 4.2. Human Cardiac Muscle Patches (hCMP), and iPSC-Derived Cardiomyocytes

hCMPs have been found to be an alternate and more effective method for cell delivery by addressing the limitations of low engraftment listed above. hCMPs are commonly of interest to treat the myocardium by increasing its regenerative capacities and preventing post-injury cardiac remodeling in the left ventricle (molecular and interstitial changes that leads to changes in size, mass, shape, and function of the heart, associated with ventricular dysfunction and malignant arrhythmias) [76,77]. A mixture of smooth muscle cells, cardiac fibroblasts, endothelial cells, cardiomyocytes, and others are used in the creation of hCMP patches, but this paper explores the effects of iPSC-derived cardiomyocytes on hCMP patches.

Gao et al. conducted a study by utilizing iPSC-derived cardiomyocytes (mentioned above), which were seeded into a 3D fibrin scaffold and cultured with dynamic and mechanical stimulation (for maturation), to create an hCMP (4 cm × 2 cm × 1.25 mm) [78]. A linear array system of electrical sensors was used to evaluate intercellular coupling, and the hCMP gap junction communication was found to be similar to that of a native rabbit myocardium. Moreover, swine models (important to note that animal models may not accurately correlate with human models) implanted with the hCMP had considerably advantageous regional wall stress, left-ventricular ejection fraction, and infarct size than a swine without the hCMP [78].

Other studies reported that hCMPs created with iPSC-derived cardiomyocytes and then implanted in rodents had higher cell engraftment rates and lower infarction size compared to iPS cell injection [79,80,81].

#### 4.2.1. iPS-Derived Cardiomyocytes and Cell Sheets

Cell sheets have been developed for several applications; for example, non-stem cell sheets for corneal pleural defects and ischemic myocardium treatment, and mesenchymal stem cells for periodontal tissue, bone defects, and myocardial treatment. In addition to stacking microstructured mesh sheets, stacking monolayers of cell sheets–created using a special culture dish coated with poly N-isopropyl acrylamide–on top of one another can create a 3D hCMP (Figure 4). The immunogenicity of the cell sheet is not a problem because the cell sheets lack synthetic scaffolding. On the one hand, the cells in adjacent layers can form connections such as gap-junctions; on the other hand, transplanted cell sheets are electrically isolated from the rest of the myocardium and use paracrine mechanisms [82,83]. Another added benefit of cell sheets is that they allow the final product to maintain adhesion proteins on its surface, allowing for easy integration when transplanted [76].

On 16 May 2018, Japan’s health ministry gave the green light for Yoshiki Sawa’s team at Osaka University to graft cell sheets created with iPSC-derived cardiomyocytes onto the human heart. Three years later, the study showed that in the 13 patients (with left ventricular ejection fraction <35%) that received the treatment, the survival rate was 90.9 ± 8.7% after 5 years, which was higher than the estimated survival rate (found with the Seattle Heart Failure Model) of 70.9 ± 5.4% [84]. This clinical trial showed the safety of said cell sheets, but more research with larger test groups must be conducted.

#### 4.2.2. Cell-Free Cardiac Patches

Scientists noticed that the mechanism of regeneration in the heart relies on the paracrine signals produced by cells, not the cells themselves [85]. This paracrine activity can be emulated with extracellular vesicles, which induce recovery by delivering miRNAs [5]. Liu et al. developed a system for the extracellular vesicles secreted from iPSC-derived cardiomyocytes to be delivered to the infarcted area, resulting in the promotion of regeneration and reduced infarct size [86].

#### 4.2.3. Spheroids and iPSC-Derived Cardiomyocytes

Spheroids describe 3D cell aggregates, which have been studied in disease modeling due to the existence of: (1) cells exposed to the surface and cells deeply buried in the structure. (2) A hypoxic center with a well-oxygenated outer layer of cells. (3) A collection of proliferating and non-proliferating cells [87]. Cells have been found to aggregate into said spheroids during iPSC-derived cardiomyocyte differentiation [88,89]. These spheroids can promote increased regeneration and function expression in the myocardium compared to 2D cultures when they are seeded into metal devices, organized, and then extracted [89].

#### 4.2.4. 3D Bioprinting and iPSC-Derived Cardiomyocytes

3D bioprinting, a technology with the ability to create highly organized vascular networks, utilize a blend bioink composed of gelatin methacryloyl and 4-arm poly-tetra-acrylate, as well as CAD modeling, to create the structures [90].

Lee et al. printed a support material to fill with cardiomyocytes at a concentration of 300 million cells/mm [91]. Kupfer et al. used Lee’s study as the inspiration to develop a bioink that promoted hiPSC viability and differentiation in a two-chambered, closed organoid. The team’s knowledge of extracellular matrices in which the proper function of cells is dependent on the spatial and temporal engagement of the matrix with cell surface integrins, as well as the successful transfer of the mass-produced hiPSC cardiomyocytes from the dish to the structure (the process required removal of the aggregated cardiomyocytes and disruption of cell connections), proved successful. The hiPSCs differentiated into mature cardiac cells, enabling the structure to mimic the function of a native heart [92].

#### 4.2.5. Transplantation of the hCMP

Cell-free cardiac patches that rely on the paracrine system, described above, can be directly injected into the heart [5]. Direct injections are non-invasive solutions; however, a large clinical study in 5 European countries with 78 participants testing injectable hydrogels in patients with chronic heart failure reported that the group that received the treatment had 3 deaths in 30 days compared to 0 deaths in the control treatment group [93]. More research must be conducted for this transplantation method to be clinically applicable.

The invasive delivery method of attaching hCMPs has several limitations, including that open heart surgery must be conducted by trained staff, which limits the possibility of repeated applications (unlike non-invasive methods).

### 4.3. Disease Modeling

Disease modeling with iPSC-derived cardiomyocytes allows for the development and evaluation of new treatments, enabling safety and efficacy compared to previous disease models with immortalized cell lines and animals [94]. An example of the unreliability of animal models is trisomy 21, for which a mouse model of this disease failed to emulate the human cranial abnormalities on human chromosome 21 or how the orthologous segments on human chromosome 21 can be found in mouse chromosomes 10 and 17 [95]. A study by Stilitano et al. showed that patients’ specific response to the drug sotalol could be predicted by analyzing iPSC-derived cardiomyocytes created from the patients themselves; the in-vitro cardiomyocyte response to sotalol was very similar to the in-vivo heart response [96].

iPS cells are drawing extra attention due to the possibility of patient-specific disease modeling and testing important factors for curing rare diseases [97]. In 2008, Park et al. were the first researchers to create disease (genetic diseases with Mendelian and complex inheritance)-specific iPSC-derived cardiomyocyte lines in culture [98,99]. Said specific cell lines from each patient could be used to create specialized iPSC-derived cardiomyocytes, which could then be analyzed.

Increased research progress in gene editing technology using clustered regularly interspaced short palindromic repeats (CRISPR) has allowed scientists to compare drug responses and disease phenotypes between patient iPSC-derived cardiomyocytes and wild type-cardiomyocytes or to create isogenic iPSC-derived cardiomyocytes for comparison [100,101,102,103].

An increasingly researched area in drug cardiotoxicity testing and drug development revolves around the use of iPSC-derived cardiomyocytes in “heart-on-a-chip” biomimicry technology. In this technique, an in vivo-like microenvironment of the myocardium is created using human iPSC-derived cardiomyocytes. Perfusion is integrated into the model to mimic dynamic extracellular conditions, achieving high similarity to the physiological environment of the myocardium [104]. Heart-on-a-chip technology has been used in 2D monolayer contexts along with, despite the name, 3D microfluidic platforms [105]. Introducing capillary permeability to heart-on-a-chip models has been challenging, with only preliminary findings exhibiting successful recreation of a permeable endothelium-lined vascular layer [104,106]. Not only have heart-on-a-chip models been used to test drug efficacy and toxicity in the heart, but they have also been used to investigate previously difficult-to-explore systems of the body. Notably, Bernardin et al. explored the neuro-cardiac junction using a heart-on-a-chip model [107]. The challenges of current heart-on-a-chip technology include low scalability, high expense, and the necessity to culture in biocompatible materials [108].

To no surprise, iPSC-derived cardiomyocytes excel in disease modeling of cardiomyopathies, i.e., heterogenous heart diseases correlated with abnormal myocardium structure and function. Cardiomyopathies are classified as follows, based on the structural and functional changes they create: arrhythmogenic cardiomyopathy (AC) (when the ventricular myocardium is continuously replaced with scar fibrosis and fat, negatively affecting the electrical signal transmissions in the heart and leading to arrhythmias); dilated cardiomyopathy (DCM) (dilation and elongation of the ventricles that could lead to heart failure, and as of 2022, treatment usually entails a heart transplant); hypertrophic cardiomyopathy; and restrictive cardiomyopathy [109,110,111].

This section explores the different prospects and factors of disease modeling with iPSC-derived cardiomyocytes as well as its setbacks. We focus on AC and DCM due to increased attention on those cardiomyopathies over the last few years.

#### 4.3.1. Disease Modeling for AC Treatments

Mutations in genes that regulate Ca+, ion channels, and desmosomes lead to AC, in which (as mentioned above) focal fatty infiltration and even the loss of cardiomyocytes occur [112,113]. Said genes found using epidemiological data in studies by Awad et al., Corrado et al., Chen et al., Padron-Barthe et al., and Thiene et al. include desmosomal genes PKP2 (plakophilin-2), DSP (desmoplakin), DSG-2 (desmoglein-2), and DSC2 (desmocollin-2) as well as non-desmosomal genes DES (desmin), CDH2 (cadherin-2), and LMNA (lamin A/C) [114,115].

Inoue et al. generated isogenic iPSC-derived cardiomyocytes and used CRISPR to perfectly adjust the expression of PKP2. The team accurately modeled AC, in which desmoglein-2 immunostaining revealed that a one-half dose reduction of PKP2 led to a reduction of desmosome assembly [116].

Disease models for AC can also be created when PKP2, or other genes associated with AC, is induced with a cell culture containing IBMX and a PPARy agonist. Kim et al. successfully studied AC using this method, inducing AC phenotypes of cell death/apoptosis [117].

#### 4.3.2. Disease Modeling for DCM Treatments

Genes causing DCM include TNNT2, MYBPC3, MYH7, and TTN, resulting in enlarged ventricles, thinning walls, decreased contractile function, and ventricular dilation [100,118,119,120].

One example of DCM disease modeling is the recent study conducted by Dai et al. in which the sarcomeric mutation in the TNNT2 gene was analyzed [121]. The team created patient-specific iPSC-derived cardiomyocytes using CRISPR and compared them to healthy TNNT2 genes. The TNNT2 gene mutation disrupted troponin’s interactions with other sarcomere proteins and also disrupted AMPK-dependent sarcomere-cytoskeleton interactions. The addition, the AMPK activator A-769662 led to an increase in contractility [121].

### 4.4. Limitations in iPSCs in Clinical Settings

#### 4.4.1. Transplantation Rejection

Immune rejection of transplanted IPS-derived cardiomyocytes, especially in allogeneic cell transplantation, is a major obstacle for clinical studies. Creating and developing iPSC-derived cardiomyocytes for each patient would be too costly and time-consuming, reducing the accessibility of the treatment. Although Kawamura et al. demonstrated in a monkey-based allogeneic experiment that immunosuppressants demonstrated fair engraftment of iPSC-CMs, renal failure, malignancy, and infection can arise from immunosuppressive agents [122,123]. Instead, Yoshida et al. found that mesenchymal stem cells reduced immune rejection and increased cell survival rates in mice by ~30% compared to the control group [124]. Moreover, transplantation from major histocompatibility complex (MHC) type-matched iPSCs can mitigate immune rejection. In a study by Shiba et al., iPSCs from MHC-homozygous animals differentiated into cardiomyocytes and survived without the help of immunosuppressive agents [125].

#### 4.4.2. Arrhythmias with Cardiomyocyte Cell Sheets and Transplantation

First, the thickness of hCMP patches is a major obstacle, for thin patches (<2 mm) with large surface areas are not able to effectively mimic the native myocardium [126]. The diffusion of oxygen and nutrients limit the thickness but an alternative to avoid this issue is to stack layers of cell sheets, thus creating an hCMP of any thickness with the ability to electrically connect between the layers [76].

Second, the occurrence of arrhythmias is prevalent in hCMP studies. Especially with electro-stimulation of the cardiomyocytes for maturation purposes, the frequency of arrhythmias increases due to changes in electrical coupling. Focusing more on cardiomyocytes, arrhythmias in cell sheets with transplanted iPSC-derived cardiomyocytes present a complication for the translation of human iPSC-derived cardiomyocytes into clinical use. Large animal models are being used to study arrhythmias because their sinusoidal waves accurately represent the physiology of the human heart. In studies with large animals, results have been mixed [76]. In pigs, ECS transplantation has been proven to be a successful approach, but ventricular tachyarrhythmias were induced in 2 out of 7 pigs, ultimately proving to be fatal [127]. However, in nonhuman primates, non-fatal arrhythmias continued for two weeks after transplantation, and declined from there on. This confusion necessitates more research into the impaired impulses at host-transplanted cardiomyocyte interfaces before translation to widespread clinical trials. Attempts to reduce arrhythmias in the context of transplanted cardiomyocytes have been studied. Two leading attempts include the use of “nodal-like” cell types and increasing the maturity of transplanted cardiomyocytes along with the use of antiarrhythmic drugs [128,129]. Antiarrhythmic drugs, in particular, are cause for concern as this class of drug is prone to over or under treatment in clinical settings, their use is outdated, classification is simplified, and the drugs often underperform compared to interventional therapy [130].

#### 4.4.3. Disease Modeling Challenges

It is worth noting that the drugs developed for AC and DCM and tested in iPSC-derived cardiomyocyte disease modeling have never been used in a clinical setting. Part of the reason is the difficulty of recapitulating a disease with multiple genetic and environmental factors, such as congenital heart disease. Microfluidics and disease-on-chip technology may also hold potential for addressing the drawbacks of iPS disease modeling, but more research is required [98].

Another challenge for disease modeling not limited to cardiomyocytes is that female-derived iPS cells have a problem with X chromosome erosion after periods of long culture. Due to the lack of available and testable brain tissue, hiPSCs are exciting for Autism Spectrum Disorder (ASD) research, but X chromosome erosion must be solved for more ASD studies. X chromosome erosion affects protein expression levels, leading to inaccurate testing with hiPSCs. During long periods of culture, X chromosome erosion changes the inactive state of X chromosomes with H3K27me3, whilst having XACT coat the eroded areas. Thus, X chromosome erosion causes abnormal gene expression [131].

#### 4.4.4. Immaturation

Additionally, the clinical use of human iPSC-derived cardiomyocytes is stymied by the differentiation and maturation methods used today, which generate ‘immature’ embryonic or fetal cardiomyocytes [132]. These immature cardiomyocytes have major structural and functional differences from adult cardiomyocytes, including a 30-fold smaller size, weaker contractile force, low conduction velocity, and a less developed electrophysiological system [133,134,135]. Although maturation solutions are being developed, these are often costly or resource inefficient. The foremost solutions include: (1) culturing cardiomyocytes for longer periods of time, which is a labor-intensive process, possibly leads to X chromosome erosion, and is understudied at the long lengths of time required to form mature adult cells [136]. (2) The use of hydrogels in 3D tissue engineering to simulate the native cardiac tissue microenvironment, which is increasingly popular and has seen great success but suffers from a severe lack of scalability [137,138]. (3) The use of thyroid hormones as biochemical cues to modulate key cardiac development proteins such as myosin heavy chain, triiodothyronine, and titin, which is a promising method [139,140,141].

#### 4.4.5. Primed iPSCs Challenges

It is important to note that there are two types of pluripotent stem cells into which to differentiate: primed and naive. After reprogramming the naive, or ground, hiPSCs, the protocols create primed hiPSCs. Similar to primed EpiSC mouse cells being unable to contribute to blastocyst chimera formation and are dependent on activin and FGF signaling, primed hiPSCs represent a restricted state that cannot be used for post-implantation [142]. Thus, many researchers have created protocols to create and then maintain naive pluripotency, such as Guo et al., Smith et al., and Vallot et al. [143,144]. More research is necessary to understand the differences between the two states, such as DNA hypomethylation and X chromosome reactivation demonstrated in naive hiPSCs, in order to successfully achieve reprogramming from the primed to naive state [142].

## 5. Ethical Considerations and Regulation Challenges of iPSC-Derived Cardiomyocytes

Although the discovery of human iPSC-derived cardiomyocytes was in 2006, ethical dilemmas still exist regarding the usage, regulation, and standardization of human iPSC-derived cardiomyocytes and iPSCs as a whole. At the heart of all issues lies the “bench to bedside” experimentation process in which iPSCs currently are being tested; a lack of regulation and ethical clarity clouds the current iPSC clinical research sphere. To date, insufficient completed clinical trials involving the use of iPSC-derived cells have been completed. In 2014, the RIKEN institute treated a singular patient suffering from age-related macular degeneration [145]. In spite of “no serious adverse effects”, the trial was halted in favor of the usage of partially matched donor cells. In the case of clinical trials involving the use of human-derived iPSC cardiomyocytes, there is a severe lack of data and testing. Through the transition from dish to bench to bedside, uncertainty remains in the transition to bedside; clinical testing and the lack of clinical trials further heightens the issue as there is a lack of precedence and every step toward testing in the clinic would be unknown. This issue remains so pervasive that McNeish et al. published a paper in 2015 detailing the path his team took when navigating a clinical trial researching the phenotypes of human iPSC-derived motor neurons in amyotrophic lateral sclerosis patients [146]. Additionally, while iPSC technology holds great possibilities for the future of stem cell research, King emphasized that ethical caution must still be present throughout the research process, from preclinical research to clinical trials and commercialization [147].

### 5.1. Human Embryos in hESC Research vs. iPSC Research

Upon the inception of the first human ESC (hESC) in 1998, ethical concerns were immediately raised. While several relatively extraneous concerns were brought up, such as human cloning, immortality, and the creation of chimeras, the pro-life movement seeks to obstruct embryo destruction—an essential part of hESC research [148,149]. Although Hyun noted several major religions (Judaism, Islam, Hinduism, and Buddhism) and many Western Christian views traditionally consider the moral beginnings of a human later in the gestation process, governments have taken stances against hESC [150]. The fundamental issue taken by the public and governments against hESCs is the destruction of early human embryos. Given the divisiveness of this issue, it is unlikely in the short term that a resolution will dictate whether the destruction of embryos is moral [151]. This has led governments to enact various levels of regulation for hESC research, from permissible regulation in the UK, to intermediate regulations in the US, to lightly monitored research in Japan, to heavy restrictions in Germany [152,153,154]. It is important to note that ethical concerns are not the only factor in regulation since the pluripotency of hESCs can lead to undesired cell differentiation in vivo and teratomas, with one study finding a teratoma appearance rate of up to 100% in immunodeficient mice [155,156,157]. In contrast to hESCs, iPSC-derived cells do not require the destruction of an embryo, making them morally preferred over hESCs [158]. In countries such as Germany or Italy with heavy restrictions on hESC research, iPSC research is this able to take place in both observational and interventional capacities [50,154].

### 5.2. Human iPSC Research Regulation

Regulation is of the utmost importance in human iPSC research, as the future possibilities and uses of human iPSC-derived cells are limitless. In the case of human iPSC-derived cardiomyocytes, regulation is necessary to ensure solutions and applications are effective and administered accordingly. Regulation is even more important today when the push to test human iPSC-derived cardiomyocytes in patients and commercialize applications is far greater than moral status or past issues with the technology [159]. Currently, a large part of the academic work on iPSC-derived cardiomyocytes is in clinical trial testing, a shift toward the “bedside” phase of research [51]. Thus it is necessary to understand the nature of clinical trial regulation in order to understand the regulation of iPSCs as a whole. There have been international efforts to consolidate the regulation of iPSCs and create a universal system of oversight. One such nonprofit is the Global Alliance for iPSC Therapies (GAiT), which primarily centralizes cell line creation, ensures ethical responsibility during donor selection and quality control processes, and regulates institutions through Good Tissue Practice and Good Manufacturing Practice compliance systems [160]. Another effort is the European General Data Protection Regulation, which in the context of iPSCs, protects the identity of tissue donors in biobanks [161]. Both regulatory bodies stand to improve the ethical standing of iPSC manufacturers.

#### 5.2.1. Regulation in Japan: A Unique Example

Regulations and support of iPSC research further vary by country, the country’s governments, stance on iPSC technology, and drug regulatory body. One notable country at the forefront of iPSC technology through lessening regulations and increased support to researchers is Japan. Recently, Japan has made steps to simplify and ease regulations to attract more iPSC-related research and commercial testing [162]. A network supervised by the Japan Agency for Medical Research and Development is leading this effort by changing laws and increasing support for iPSC research [163]. Additionally, Japan stands out among many other countries for its continued support of iPSC research, especially after Yamanaka’s sharing of the 2012 Nobel Prize in Physiology or Medicine. Japan uniquely encourages collaboration between the government and research institutions, further accelerating research. Japan stands out as a government in support of iPSC research [163].

#### 5.2.2. Nomenclature Outside of Academia

Lack of singular, clear, and internationally recognized regulation affects non-scientific and academic interpretations of the term “stem cell”. Broadly, a stem cell may refer to an iPSC, hESC, multipotent cell, totipotent cell, or hematopoietic stem cell (HSC), which have long been used in stem cell therapy, unlike experimental iPSC and hESC therapies. Indeed, in reference to the Dutch Embryos Act and European Convention on Human Rights and Biomedicine, one article pointed out the vague and inconsistent terminology when referring to hESC and iPSC technology [164]. Journalists, the general public, and, most importantly, patients may find ambiguity between the terms used in regulation laws and unproven stem cell therapies regarding iPSCs [150,159,165,166,167,168].

## 6. Conclusions

The benefits of iPSC-derived cardiomyocytes in applications of tissue engineering and disease modeling are apparent. In this paper, we provided multiple studies to demonstrate this new field of research and we discussed differentiation, enhancements, applications, and ethics relating to iPSCs. The main obstacle regarding the area of research is the extremely large number of differentiation and reprogramming protocols with similar results, thereby making uniformity hard to achieve and therefore limiting the number of clinical trials. Moreover, we found it hard to find up-to-date, reliable research from the last three years in specific research areas, such as differentiation. Those areas of research must be further pursued, which may occur soon as research regulations loosen.

## Figures and Tables

**Figure 1 bioengineering-10-00237-f001:**
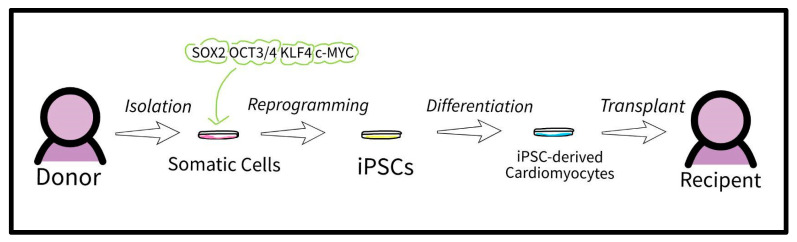
The process from isolation to transplantation for an iPSC-derived cardiomyocyte [15].

**Figure 2 bioengineering-10-00237-f002:**
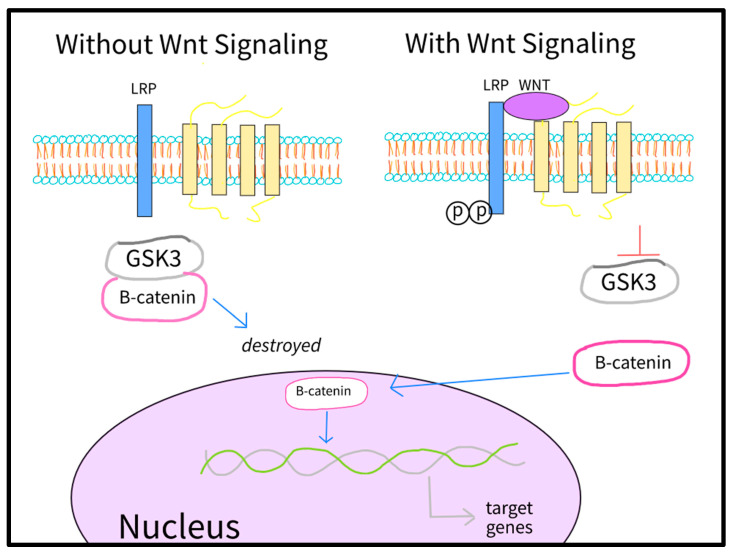
Inhibition of GSK3 in Wnt signaling leads to activation of β-catenin and transcription of differentiation genes [37].

**Figure 3 bioengineering-10-00237-f003:**
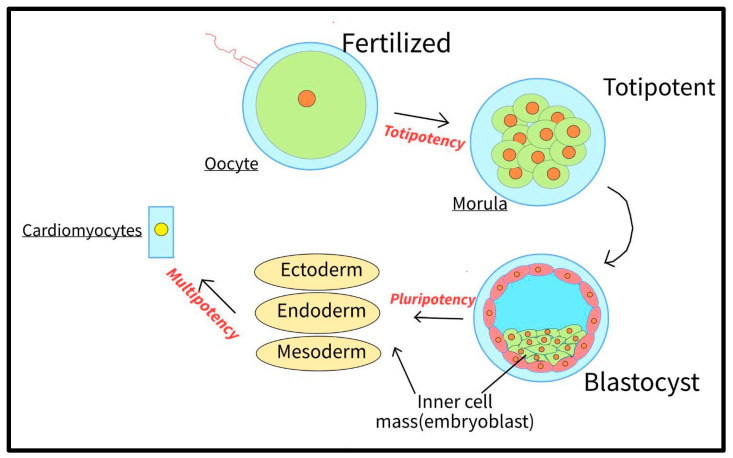
Diagram showcasing the steps from a fertilized egg to a cardiomyocyte; totipotency, pluripotency, and multipotency achieved through the process for embryonic stem cells.

**Figure 4 bioengineering-10-00237-f004:**
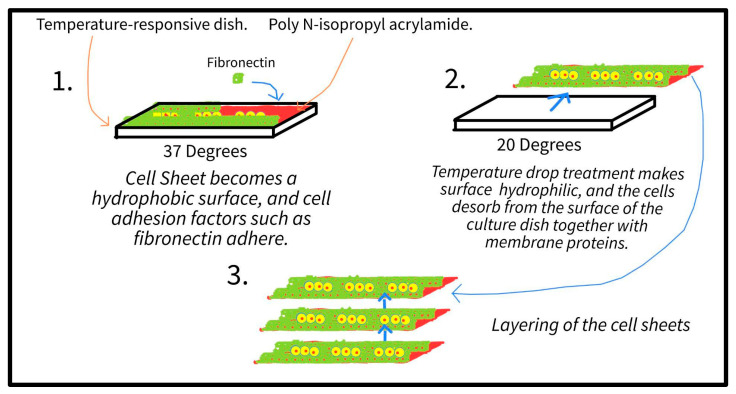
A more detailed explanation for the creation of cell sheets. Step 1 demonstrates the culture of the iPS-derived cardiomyocytes, step 2 describes the creation of each individual cell sheet, and step 3 describes the layering of the cell sheets the result in the final creation [65].

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
