# Peer review of "The Exciting Realities and Possibilities of iPS-Derived Cardiomyocytes"

_bioengineering, 2023, doi:10.3390/bioengineering10020237_

Round 1

Reviewer 1 Report

1-     What is QT syndrome? All abbreviations need to be defined in whole manuscript carefully

2-     Some sections need proper references: for instance

“One of the most promising differentiation routes includes human iPSC-derived car- 39

diomyocytes. The ability to create cardiomyocyte tissue, which induces contractile motion 40

in the heart, opens new avenues of regenerative research such as transplantation, bioprint- 41

ing, disease modeling, and cardiac patches.”

3-     I suggest discussing the following paper in sections related to scar formation in myocardial infarction and role of type V collagen in regulating scar size : https://www.sciencedirect.com/science/article/pii/S0092867420308072

4-     For the applications section, it is suggested to discuss using ips-derived CMs in organ on a chip and micropysiological systems for disease modeling and drug screening

5-     Challenges and future direction can be discussed further

Author Response

Responses to Editorial requests and Reviewers’ comments

We greatly appreciate the editorial requests and suggestions. We believe that the manuscript has been significantly improved by these revisions. Below are our point-by-point responses to the editorial requests and reviewers’ comments. Changes made in response to the referees’ comments are shown in red in the revised manuscript and in this letter. The text in blue in this letter indicates the editorial requests and reviewers’ comments.

Response to Reviewer #1

Thank you for the very useful and interesting advice. The responses to your comments are described below.

Reviewer 2 Report

1. The introduction is started US data for cardiovascular diseases, it is recommended to write the global figure for CVD. 

2. Figures images and writing is too small to read and understand, it is recommended to improve quality of figure. 

3. Tables should be introduced separately for small molecules and genetic modifications used for the differentiation of IPSCs to cardiac cells. Small molecules, inhibitors and signaling activators, drugs etc. must be combine under a single heading.

4. Limitation for the use IPSCs in clinical and research setting must be added.

5. The clinical studies conducted so far for the use of IPSCs in CVD, must be added.   

6. The last heading “Discussion and Summary” should be removed or more details should be added, so that the section is complete and comprehensive.

Author Response

Responses to Editorial requests and Reviewers’ comments

We greatly appreciate the editorial requests and suggestions. We believe that the manuscript has been significantly improved by these revisions. Below are our point-by-point responses to the editorial requests and reviewers’ comments. Changes made in response to the referees’ comments are shown in red in the revised manuscript and in this letter. The text in blue in this letter indicates the editorial requests and reviewers’ comments.

Response to Reviewer #2

Thank you for the very helpful advice. The responses to your comments are described below.

Reviewer 3 Report

The authors provide a literature review of the differentiation process of human iPS cell-derived cardiomyocytes, its potential application and ethical issues.

The description of differentiation methods seems inadequate. Even in the application section, there is insufficient description of important issues such as arrhythmias and rejection. In addition, some reference literatures and the texts do not match.

The following are specific comments.

Line 30-32

“Other treatments include pharmacological treatments such as beta blockers, angiotensin receptor blockers, I(f) inhibitors, and more-but fail to have drastic improvements in patients.”

Early reperfusion therapy and pharmacological treatments as authors described restore cardiac function in patients with acute myocardial infarction, and these pharmacological treatments often restore cardiac function dramatically in patients with dilated cardiomyopathy. Therefore, I think authors should correct this sentence.

Line 67

GSK3 is almost synonymous with Wnt signaling?

Line 96

Knockout serum replacement is a defined composition. If albumin is undefined, then B-27 supplement is also undefined because B-27 supplement contains albumin. There are also many cases of EB differentiation using defined media. Please review the literature carefully and correct this section.

Line 100-103.

Authors first listed Wnt inhibitors. But they should start with Wnt activators, CHIR, BIO and anything?

Line 103

There are reports of Wnt-C59 being used as a Wnt inhibitor. Please add it.

Line 106-107

EB differentiation can also induce differentiation by regulating Wnt signaling (Kempf H et al. Stem Cell Rep. 2014). Please correct.

Line 111

EB differentiation (suspension culture) can also induce cardiomyocyte differentiation with high efficiency if it is optimized (Kempf H et al. Stem Cell Rep. 2014). Even in monolayer culture, if cell density, GSK3 inhibitor concentration and culture period are not optimized, some cell lines cannot induce cardiomyocytes efficiently. Please correct.

Line 121

Please cite the appropriate reference, as the text and the reference [30] do not match.

Line 123-128

The text and the reference do not match.

Line 132-134

“Liam” is wrong. The text and the reference do not match.

Line 143-144

“The timing of the Wnt signaling” is strange. Please correct it.

Line 147-149

The protocol using GSK3 inhibitor and Wnt inhibitor differentiates even with insulin; Lian X et al. Nat Protoc. 2013 shows a protocol for adding insulin to some cell lines. Please correct.

Line 167-168

Pluripotency refers to the ability to differentiate into all three germ layers. Please correct.

Line 174-175

The sentence is not correct. Please correct.

Line 177-179

Chemical reprogramming has been reported in Nature. 2022 May;605(7909):325-331.

Line 184-185

What are “the third and fourth methods”? I don't understand.

Line 211-217

I don't understand whether authors want to talk about cardiomyocytes or pluripotency. Please correct the sentence.

Line 218

There must be many other ways for cardiomyocytes to mature. Please add the following: increase in fatty acid metabolism, electrical stimulation, epigenetic priming, etc.

Line 271-279

The article in reference [67] does not examine the use of pigs. Please cite the appropriate reference.

When cell sheets are implanted on the epicardial side, do they bind electrically to the host myocardium? If not, it would not be able to contribute to contraction. The human heart, unlike the pig heart, is covered with fat. Please provide literature and thoughts on this point.

Line 284-291

Please provide a summary of the many methods of making cardiac sheets using other materials that have been reported.

Line 350

There is concern that ventricular arrhythmias may be a problem with cardiomyocyte transplantation. Although not a cell sheet, attempts to reduce arrhythmias caused by transplanted cardiomyocytes have been reported, including modulation of cell types and maturity of transplanted cardiomyocytes (Ichimura H et al. Sci rep. 2020; Nakamura K et al. Stem Cell Reports. 2021) and use of antiarrhythmic drugs.

Other comments

l  The most problematic aspect of cell transplantation, rejection, and challenges to solve this problem are not mentioned at all. Please address this issue.

l  There are naive and prime types of human iPS cells, and there are differences in their pluripotency maintenance mechanism, proliferative ability, and differentiation ability. Please address this.

l  Female-derived iPS cells have a problem of X-chromosome inactivation disruption associated with long-term culture. This seems to affect differentiation potential and expression levels of proteins derived from sources other than the X chromosome. This affects both transplantation and disease research, so this should be mentioned as well.

l  The type of cardiomyocytes induced is an important factor for both transplantation and disease models. Authors should discuss it in the section describing differentiation processes.

Author Response

Responses to Editorial requests and Reviewers’ comments

We greatly appreciate the editorial requests and suggestions. We believe that the manuscript has been significantly improved by these revisions. Below are our point-by-point responses to the editorial requests and reviewers’ comments. Changes made in response to the referees’ comments are shown in red in the revised manuscript and in this letter. The text in blue in this letter indicates the editorial requests and reviewers’ comments.

Response to Reviewer #3

Thank you for your insightful comments and advice. The responses to your comments are described below.

Reviewer 4 Report

This review is well written, covering a wide variety of topics regarding human iPSC-derived cardiomyocytes in both basal and clinical points of view. Since this manuscript provides an excellent systemic review, it will even serve as a textbook for students, young researchers and clinicians. If authors add a description regarding topics about iPSC-derived pacemaker cells, the content of the manuscript will be further attractive. 

Minor concerns:

1) In line 365, the word “patent-specific” should be corrected as “patient-specific”.

2) In line 366, the phrase "testing-an important factor ...." should be corrected as "testing an important factor ...".

Author Response

Responses to Editorial requests and Reviewers’ comments

We greatly appreciate the editorial requests and suggestions. We believe that the manuscript has been significantly improved by these revisions. Below are our point-by-point responses to the editorial requests and reviewers’ comments. Changes made in response to the referees’ comments are shown in red in the revised manuscript and in this letter. The text in blue in this letter indicates the editorial requests and reviewers’ comments.

Response to Reviewer #4

Thank you for the advice and help with proofreading. The responses to your comments are described below.

Round 2

Reviewer 2 Report

The concerns raised in the first round of review are addressed in the revised version. 

The manuscript is accepted in its current form. 

Reviewer 3 Report

Authors have corrected in response to the points I pointed out. No particular comments.